# Controlling the Quality of Nanodrugs According to Their New Property—Radiothermal Emission

**DOI:** 10.3390/pharmaceutics16020180

**Published:** 2024-01-26

**Authors:** Gleb V. Petrov, Daria A. Galkina, Alena M. Koldina, Tatiana V. Grebennikova, Olesya V. Eliseeva, Yana Yu. Chernoryzh, Varvara V. Lebedeva, Anton V. Syroeshkin

**Affiliations:** 1Department of Pharmaceutical and Toxicological Chemistry, Medical Institute, RUDN University, 6 Miklukho-Maklaya Street, 117198 Moscow, Russia; 2Federal Government Budgetary Institution “National Research Center for Epidemiology and Microbiology Named after Honorary Academician N.F. Gamaleya” of the Ministry of Health of the Russian Federation, 18 Gamaleya St., 123098 Moscow, Russia

**Keywords:** nanodrugs, interferon-α2b, interferon-β, interferon-γ, VLP vaccines, virus-like particles, SARS CoV-2, radiothermal emission, quality control, nanoparticles

## Abstract

Previous studies have shown that complexly shaped nanoparticles (NPs) have their intrinsic radiothermal emission in the millimeter range. This article presents a method for controlling the quality of nanodrugs—immunobiological preparations (IBPs)—based on the detection of their intrinsic radiothermal emissions. The emissivity of interferon (IFN) medicals, determined without opening the primary package, is as follows (µW/m^2^): IFN-α2b—80 ± 9 (10^5^ IU per package), IFN-β1a—40 ± 5 (24 × 10^6^ IU per package), IFN-γ—30 ± 4 (10^5^ IU per package). The emissivity of virus-like particles (VLP), determined using vaccines Gam-VLP-multivac (120 μg) in an injection bottle (crimp cap vials), was as follows: 12 ± 1 µW/m^2^, Gam-VLP—rota vaccines—9 ± 1 µW/m^2^. This study shows the reproducibility of emissivity over the course of a year, subject to the storage conditions of the immunobiological products. It has been shown that accelerated aging and a longer shelf life are accompanied by the coagulation of active NPs, and lead to a manyfold drop in emissivity. The dependence of radiothermal emission on temperature has a complex, non-monotonic nature. The emission intensity depends on the form of dosage, but remains within the order of magnitude for IFN-α2b for intranasal aqueous solution, ointments, and suppositories. The possibility of the remote quantitative control of the first phases of the immune response (increased synthesis of IFNs) to the intranasal administration of VLP vaccines has been demonstrated in experimental animals.

## 1. Introduction

This Special Issue was conceived by the editors to publish fundamental and practical results on the study of the characteristics of drugs based on NPs: this is the subject of a kind of editorial article [1]. The idea of the Guest Editors was to reveal a fundamental property of biologically active NPs. In contrast to the classical multi-point Coulomb interaction of a ligand with a receptor [2,3,4], the binding of a NP to a receptor involves a significant contribution from the Deryagin interaction—the mutual adjustment of flickering dipoles during the electromagnetic radiation of sticking surfaces [5,6]. As it turns out, if a biologically active NP has a non-spherical shape (non-convex polyhedron), then the formation of an unstable dipole with electron plasma-like inhomogeneity is possible [1]. In this case, the NP acts as a kind of “nano antenna-emitter”, the function of which, when a “nano antenna-receiver” appears, is to adjust the frequency and localization of the flickering dipoles to the energetically optimal adhesion of the surfaces [1,7,8]. In the general case, the radiation of a nanoantenna should be isotropic or with weakly expressed anisotropy [9,10]. The isotropy of radiation from “medicinal” NPs opens up for pharmaceutical chemistry an amazing opportunity to control the presence of NPs and their concentration in finished dosage forms and in the human body, including without opening the packaging [11]. A new chemical–analytical method for the remote monitoring of the quality of medicines was created.

In this paper, the following finished dosage forms were chosen for study: VLPs and IFNs. VLP formation can occur either naturally or synthetically, due to the individual expression of viral structural proteins, forming a virus-like structure during self-assembly without nucleic acids, which ensures the optimal presentation of the antigen in the vaccine [12,13]. In VLP, there are conformational viral epitopes that cause a strong T- and B-cell immune response [14]. The radiuses of VLPs range from 20 to 200 nm [1,15], which provides the best efficiency for penetration into the body’s lymphoid system. Recent experience in the development of VLP SARS-CoV-2 and rotavirus VLPs has demonstrated their high immunogenicity and their absence of side effects [16,17,18].

IFNs are a group of cytokines, which are glycoproteins that function as multifunctional broad-spectrum bioregulators [19,20]. Modern IFN medicines are genetically engineered by growing recombinant strains of *E. coli* bacteria with IFN genes, which are obtained from leukocytes or connective tissue—human fibroblasts, built into bacterial DNA [21,22]. Pharmacologically, there are two types of IFNs: type I includes groups α and β, with sub-groups α2b, β1a and β1b, and type II includes the group of IFNs-γ [23,24]. IFNs-α2b are proteins with structures of 165 amino acids, with molecular weights of 19.3 kDa [25]. IFNs-β are NPs—proteins consisting of 165 amino acids with molecular weights of 18.5 kDa. Patients with multiple sclerosis are prescribed groups β in various dosages from 8 to 24 × 10^6^ IU, depending on the age and complexity of the disease (*cursus morbi*) [26,27]. IFNs-γ is a dimerizated cytokine, the structure of which includes 143 amino acids with a molecular weight from 17 to 25 kDa. A distinctive characteristic of IFNs-γ among other IFN medicines is their use in the combination of supportive immunotherapy and anti-retroviral therapeutics for human immunodeficiency virus [28].

In this article, the millimeter emissions from finished dosage forms of IFNs and VLPs without opening the packaging were recorded. The stability of this radiothermal emission has been demonstrated throughout the year. Researchers have thoroughly studied how the flux density of radiothermal emission decreases when storage periods are violated and during artificial aging. It has been established that the decrease in the radiothermal emission of biologically active NPs during drug aging is associated with their coagulation. Bearing in mind the fact that IFNs are cytokines, it has been demonstrated that in experimental animals it is possible to remotely monitor the development of the cytokine response after vaccination using the intensity of the millimeter emission. Thus, the proposed article illustrates how the main idea of this Special Issue is being introduced into practical pharmaceutical chemistry—through controlling the stability of biologically active NPs.

## 2. Materials and Methods

### 2.1. IFN Pharmaceuticals

To carry out research in the field of intrinsic radiothermal emission, pharmaceuticals of three different groups of IFNs were selected: α–α2b; β–β1a and γ. To detect microwave emission from APIs, therapeutics with various dosage forms were used (registration numbers in the State Pharmacopoeia of the Russian Federation are indicated in brackets), and the following IFN-α2b pharmaceuticals were selected: Grippferon in the form of intranasal drops, 10^4^ IU (LP-001503), Viferon suppositories, 3 × 10^6^ IU (P N000017/01), and Viferon ointment 40 × 10^4^ IU (P N001142/01). Also included was IFN- β1a, contained in the Teberif solution for subcutaneous injection, in the amount of 24 × 10^6^ IU (LP-004137). IFN-γ in the Ingaron, as lyophilisate for preparing a solution for intranasal administration with a dosage of 10^5^ IU (LS-001330) were also included.

### 2.2. VLP Vaccines

The production and purification of recombinant VLPs formed via nucleocapsid proteins (VP2, VP6) and surface proteins VP4 or VP7 of rotavirus type A, synthesized in the baculovirus expression system, are described in [29]. Standard molecular genetic and cytological methods were used: gene synthesis; the cloning of transfer plasmids; recombinant baculoviruses’ production in Bac-to-Bac expression system; VLP production in insect cells; centrifugation in sucrose solution; enzyme-linked immunosorbent assay (ELISA); electron microscopy; polyacrylamide gel electrophoresis, and Western blot analysis.

The production and purification of recombinant VLPs consisting of four structural proteins, S, M, N, and E, of the SARS-CoV-2 virus, synthesized in the baculovirus expression system, are described in [1].

### 2.3. The Measurement of Intrinsic Radiothermal Emission

The flux density of radiothermal emission in the microwave range was determined using a TES—92 instrument (TES Electrical Electronic Corp., Taipei, Taiwan) with a sensor configured for anisotropic measurement along the Z axis. The measurement results are the maximum average flux density value with increments averaging every 300 ms. The measurements were carried out both without opening the primary package and with opening the package. The medicines were heated to 37 °C using a solid-state thermostat with Peltier elements (Termo 24-15 Biokom, Moscow, Russia). The temperature of the specimens was controlled by a remote laser infrared thermometer (Benetech GM320, Shenzhen Jumaoyuan Science And Tecchnology Co., Ltd., Shenzhen, China). To maintain the temperature range of 2–8 °C, refrigerant elements were used. In case of VLP vaccines, activation was carried out using new-generation LEDs with a power of up to 50 mW/cm^2^ in the region of 412 nm with a spectral line width of up to 2–4 nm (including model AA3528LVBS/D, type C503B-BCN-CV0Z0461, CreeLED, Durham, NC, USA) [15]. The installation geometry did not change and was strictly observed for each specimen. The spatiotemporal distribution of microwave background emission was controlled before each measurement with height, width, and length increments of 50 cm. The background emission did not exceed 1 μW/m^2^. Aqueous solutions were applied in drops of 100 μL to the center of sterile 10 cm Petri dishes. Drops were applied in the axial direction along the Z axis at a distance of 10 mm from the measuring sensor of the device. The measurement of intrinsic radiothermal emission without opening the package was carried out with the same geometry as for aqueous solutions; the primary package was placed on a stand in the form of a 40 mm Petri dish horizontally in the axial direction along the Z axis at a distance of 10 mm from the measuring sensor. Dosage forms such as ointments and suppositories were placed in the center of the Petri dish in the axial direction along the Z axis at a distance of 10 mm from the measuring sensor. The integrity of the suppositories was intact, and their weight (1.2 g) was taken as the reference weight for measuring the radiothermal emission of dosage forms. To assess the validation characteristics of in-lab precision, all measurements were performed at least seven times.

### 2.4. Nanoparticle Size Determination

The dispersed phase of the studied specimens was controlled using the Zetasizer Nano ZSP dynamic light scattering (DLS) spectrometer (MALVERN Instruments, Malvern, UK). Using this method involves analyzing diluted specimens. NP sizing is available in the range of 1 nm to 10 µm using proprietary non-invasive backscatter technology. The measurement principle is based on photon correlation spectroscopy, which is based on the analysis of the Brownian motion of particles of the dispersed phase in a dispersion medium, which leads to fluctuations in the local concentration of particles, local heterogeneities of the refractive index, and the intensity of scattered light. The characteristic relaxation time of intensity fluctuations is inversely proportional to the diffusion coefficient. Particle size was calculated using the Stokes–Einstein equation:(1)$$D=\frac{k_{B\;}T}{6\pi \eta r}$$
where *D* is the diffusion coefficient, $k_{B}$ is the Boltzmann constant, *T*—absolute temperature, η —viscosity, and r is the radius of the particle.

Each specimen was measured at least 7 times. The results were processed using the OriginPro Lab (USA) software package 21.

### 2.5. The Application of the Technique in Preclinical Trials (Measurements of the T-Cell Response)

The possibility of using the technique in the laboratory analysis of finished VLP vaccines against SARS-CoV-2 at the stage of preclinical studies was investigated. The studies were conducted using the relevant animal species of *Mesocricetus auratus*. Two groups of 4 animals in each one were formed. This scheme was reproduced three times. Twenty four hamsters (*Mesocricetus auratus*) (females, 35–45 g body weight) were purchased from Stolbovaya breeding and nursery laboratory (Research Center for Biomedical Technologies of FMBA, Stolbovaya, Moscow Region).

All the in vivo animal experiments were carried out in accordance with the Order 199n of the Ministry of Health of the Russian Federation No. 43,232 and with the “Regulations on the Ethical Attitude to Laboratory Animals of Testing Center of N.F. Gamaleya NRCEM (Moscow, Russia)”, approval code No. 04/23, approval date 27 February 2023, and according to the guidelines of the Declaration of Helsinki.

The choice of doses was based on the previously obtained results of preclinical studies conducted in coordination with the Scientific Center for the Examination of Medical Products of the Ministry of Health of the Russian Federation.

Euthanasia of animals was carried out using the method of cervical dislocation (with preliminary euthanasia with medical ether).

The first group was immunized three times with a vaccine dosage of 120 μg/dose of total protein. The vaccine was administered intranasally by instillation. The second group of animals was administered saline solution according to the scheme similar to the first group. The volume of administered material was taken into account, so it was equivalent for both groups. The intrinsic radiothermal emission was controlled on days 1, 5, 8 and 12 after each immunization. Background values in laboratory conditions were equal to ~0.1 μW/m^2^.

#### Cellular Response

The cellular response was assessed by the proliferation of lymphocytes in the lymphocyte blast transformation response (LBTR). All manipulations were performed under sterile conditions. Spleens were removed from hamsters and homogenized in 3 mL of pure RPMI-1640 medium in a sterile homogenizer. The cell suspension was centrifuged on a one-step Ficoll density gradient (PanEco, Moscow, Russia), and the fraction of mononuclear cells was isolated and washed twice in pure RPMI-1640 medium and placed in 96-well culture panels at a concentration of 10^5^ cells in 100 μL per well. A total of 100 μL per well of stimulator antigens was added to the cells for the final concentrations. Splenocytes activated with Concanavalin A (ConA, 12.5 mg, PanEco, Moscow, Russia) served as a positive control. The following cultures were used as negative controls: cell cultures from the spleens of non-immunized hamsters; unstimulated cell cultures from hamster spleens; and cultures stimulated with a nonspecific antigen—CCHF antigen (Crimean-Congo hemorrhagic fever, lat. *febris haemorrhagica crimiana*). The SARS-CoV-2 virus was used as a specific stimulant. Cells were cultured in RPMI-1640 Complete Medium with 20% fetal bovine serum (FBS), 2 mM glutamine, 4.5 g/L glucose, 50 μg/mL gentamicin, and 0.2 U/mL insulin at 37 °C in the atmosphere of 5% CO_2_. The proliferation of splenocytes was assessed in the blast transformation reaction after 4–5 days using an inverted microscope (magnification ×400). LBTR results were expressed as the proliferation stimulation index (PSI)—the ratio of the average number of blasts in the presence of stimulants to the average number of blasts in the absence of stimulants. The result was considered positive if the PSI was above 2.

### 2.6. Metrological Support

The statistical processing of the results, plotting, and the determination of averages and standard deviations was carried out using the OriginPro Lab 21 software (Origin Lab Corporation, Northampton, MA, USA).

## 3. Results and Discussion

Earlier, the possibility of detecting radiothermal emissions in the microwave range for medicines consisting of complex-shaped NPs was discovered. This discovery assumed that it is the microwave range that plays a key role in the “tuning” of flickering dipoles in the process of the dispersion interaction of the surfaces of supramolecular assemblies. The phenomenon of plasmon resonance at the interaction of these NPs with substrates should not be excluded. If two such molecules come close to a sufficiently small distance (up to 100 nm), the flickering dipoles act on each other in such a way that at each moment in time, unlike with the charges in different molecules, they are closer to each other than like ones, which causes the emergence of attractive forces. At this moment, the excitation of plasma-like regions occurs, which will emit electromagnetic waves in the microwave range. This mechanism allows for the interpreting of many phenomena related to the biological activity of NPs. In this regard, it was suggested that it would be possible to detect intrinsic radiothermal emission from biologically active NPs upon their direct activation. This was proposed to activate the emission of IFNs by heating the medicines. High-resistivity water was used as a control specimen to exclude random interference (Figure 1).

Indeed, the recorded radiothermal emission of activated pharmaceuticals (30–35 °C) was 2.5 times higher than that at 5 °C, which corresponds with the temperature regime of the storage of the therapeutics at 2–8 °C. Under storage conditions, the therapeutics also exceeded the background values (1 μW/m^2^).

### 3.1. The Radiothermal Emission of IBPs

#### 3.1.1. In-Lab Precision

An important aspect for any experimental method for therapeutic quality control is the reproducibility of the results. In this paper, we describe a year-long study of a collection of pharmaceuticals without opening the primary package. Each of the medicines had its intrinsic range of radiothermal emission (Figure 2). Based on the presented average maximum values for the annual period, it is clear that each of the studied medicines exceeds the background values (1 μW/m^2^) dozens of times, and the fluctuation of radiothermal emission from month to month varies by ±5 μW/m^2^. A Petri dish of 10cm in diameter with a drop of high-resistivity water, with a volume of 100 μL, applied strictly in the center of the dish, served as a control specimen (6).

#### 3.1.2. Specificity

IBPs are thermodynamically unstable protein structures by nature. There are special requirements for the storage of such pharmaceuticals, mainly the temperature storage conditions (2–8 °C). Also, IFNs have a shorter shelf life compared to other medicines after opening the packaging. The possibility of distinguishing IFN and VLP vaccines with the following qualitative characteristics using their intrinsic radiothermal emissions was investigated: the proper shelf life (the preparation was not opened and was produced no longer than 3 months ago); a preparation with an expired shelf life (the preparation was opened and stored for more than one and a half months under certain temperature conditions; a preparation which was not opened and was stored under certain temperature conditions for more than one and a half years); a preparation of an inadequate quality (where the preparation was artificially aged by heating it at a temperature of 50 °C for an hour; and a preparation that was stored for a month at an average temperature of 25 °C) (Table 1).

The theoretical analysis preceding the experimental data was based on the idea of changing the conformational transitions of protein structures. When the shelf life expires, the protein molecules constituting the base of medicines and IBPs lose their mobility due to a decrease in the amount of API per dose of the pharmaceuticals, due to possible protein denaturation. During accelerated aging, protein structures will coagulate due to exposure to temperatures unusual for their storage (above 42 °C), which is confirmed by the results of particle size control using the DLS method. VLPs, compared to IFNs, are more stable with respect to changes in environmental conditions, so the artificial aging technique for them differed from the technique described above—heating at 100 °C for 5 min. More stable particle-size indicators were observed compared to the situation with IFN (Table 2).

It is worth noting the dependence of the radiothermal signal on the particle size; the smaller the particle diameter is, the higher the emission detected, and vice versa. This can be explained by the same interpretation of a change in protein conformation, as well as a change in the strength of the dielectric field observed on the surface of the particles.

To determine the possibility of an unambiguous assessment of the analyte in the presence of various excipients, the intrinsic radiothermal emission was measured for three different dosage forms of IFN-α2b: solution, suppositories, and ointments for external use (Figure 3).

In liquid dosage forms, the particles are in a more mobile state compared to soft forms, due to the high viscosity of vaselinum, anhydrous lanolin in the case of ointments, and cocoa butter in the case of suppositories [30]. This leads to a diffusion obstacle: the mutual attraction of charged particles [31]. It is also worth noting that in the case of a liquid dosage form, the dispersion medium is purified water, and the excipients are sodium and potassium salts (sodium chloride, sodium hydrogen phosphate dodecahydrate, and potassium dihydrogen phosphate), which can affect improved radio wave conductivity.

To determine the sensitivity of this method, different aliquots of the studied pharmaceuticals were measured. High-resistivity water (Milli-Q) served as a control for random electromagnetic interference. When adding a volume of 1 μL of the pharmaceutical to a Petri dish, a 20-fold difference vs. the control measurement is noticeable. With an increase in the volume and concentration of medicines, the signal of radiothermal emissions from them naturally increases (Figure 4).

When studying the dependence of radiothermal emission on the specific volume and concentration of IBPs, it was revealed that with increasing the volume, the emitting activity also increased. This can be explained by the fact that as the volume of the studied IFN-α2b preparation increases, the number of charged NPs will also increase, thereby causing an increase in the number of nanoantennas emitting in the microwave range (Figure 4A). This was confirmed by investigating the emissivity of the VLP vaccine at a constant aliquot volume and a different concentration of the active substance (Figure 4B).

### 3.2. The Application of the Method in Preclinical Trials

Due to the ease of use, the method significantly accelerates the analysis of pharmaceuticals at different stages of development and production. A significant aspect of the research into the intrinsic microwave emission of IBPs and medicines is the possibility of determining the formation of the body’s immune response to the administered vaccine in vivo. In case of measuring the emission from living objects, it is worth taking into account the direction of the installation geometry (Figure 5), since adipose and epithelial tissues, as well as undercoat hair, are factors blocking the emission [1]; therefore, it is necessary to direct the detector to open areas of the animal’s body such as mucous tissue and areas of the body not covered with hair (paws, ears). The results of the measurements of intrinsic radiothermal emission of the experimental groups are shown in Figure 6.

On days 1 and 5 after the first immunization, there was a slight fluctuation in radiothermal emission results compared to the control group, which were slightly higher than the background value. A sharp spike occurs on day 8 after the first immunization, which may signal the beginning of the formation of an immune response in the bodies of the immunized animals. The second immunization resulted in a spike in radiothermal emission on day 5. The third immunization had no effect on the emissivity.

An assay of the cellular immune response to the administration of the Gam-VLP-multivac vaccine, 120 μg/dose, was carried out. The LBTR reaction was carried out on day 10 after the first, second and third immunizations (Figure 7).

The mean PSI was shown to exceed the threshold value after the first immunization with an intranasal administration of the Gam-VLP-multivac vaccine, using a 120 μg/dose (2.2 ± 0.3). After the second intranasal immunization, the average PSI value in the case of immunization with the vaccine (2.3 ± 0.4) was significantly higher than the PSI of the control group (1.3 ± 0.5), but did not increase statistically significantly when compared to similar PSI data after the first immunization. It is noteworthy that the average PSI value in the case of an intranasal administration of the Gam-VLP-multivac vaccine decreases after the third immunization (1.3 ± 0.1).

The results obtained indicate that hamsters receiving an intranasal Gam-VLP-Multivac vaccine, for the prevention of COVID-19, are potentially capable of a quantitatively effective cell-mediated immune response. Moreover, the maximum increase in a T-cell response was observed after the second immunization.

The results obtained can be interpreted as follows: the formation of an immune response in experimental subjects took from 5 to 7 days, characteristic symptoms were observed in the form of decreased activity in vaccinated individuals and decreased interest in food during these periods of time. At the moment of the formation of cell-mediated immunity, namely, the release of IFNs [32,33] in response to a foreign agent, both their intrinsic radiothermal emission and the behavior of the experimental subjects changed; the activity of individuals increased, and more aggressive behavior of the vaccinated individuals was observed. Re-vaccination led to the final formation of the IFN response and the launch of adaptive humoral immunity, and the administration of the third dose was necessary to maintain the further formation of acquired immunity [34,35,36]. In the following days before re-vaccination, a decrease in radiothermal emission was observed, which was expected, since the re-vaccination procedure was aimed at maintaining and strengthening the post-vaccination immune response due to a decrease in the activity of the immune system: the intrinsic radiothermal emission would naturally decrease, since the formation of the IFN response was near completion [37,38].

An active study of the immunogenic role of IFN and VLPs took place in the 1960s [39,40]. Research carried out in this area opened not only a new chapter in the understanding of human immune system processes, but in the also latest approaches to the treatment of viral infections and autoimmune diseases. Currently, IFNs are highly effective in the treatment of viral diseases, such as acute respiratory viral infection, influenza, herpes virus, and SARS-CoV-2 [41,42,43,44,45]. Despite the canonical role attributed to IFNs, their use is much broader than just antiviral medicines. IFN-β is widely used in supporting therapy for the treatment of the autoimmune disease MS [46,47,48], and IFN-γ is used in immunotherapy for cancers [49,50]. Present-day developments in the field of virology and genetic engineering suggest taking the experience of vaccination to a new level. The latest VLP vaccines, due to the absence of genetic material, the DNA/RNA of the virus, demonstrate greater effectiveness and safety compared to their predecessors [51]. Many VLP vaccines are currently in preclinical and clinical trials [52,53].

This study proposes a new method for the quality control of IBPs based on their intrinsic radiothermal emission. As described earlier, oppositely charged molecules at fairly short distances (to 100 nm) tend towards mutual attraction, thereby forming a dipole–dipole interaction [5,54,55,56]. Their feature is the Coulomb superstructure of flickering dipoles due to mutual electromagnetic emission being formed on the surface of NPs (including its polarization) [2]. The polarization of irregular-shaped particles leads to the appearance of a potential of up to 200 mV, which leads to the formation of an electric field strength exceeding 2 × 10^6^ V/cm and promotes the formation of quasi-plasma objects from supramolecular electrons [1,57]. These objects are capable of emitting in a wide range of frequencies, including the microwave region of the spectrum [58].

Under different temperature conditions, a natural dependence of the radiothermal emission signal of medicines on temperature is observed (Figure 1). It is believed that the “activation” of a medicine occurs when it enters the human body, while, before that moment, it is in an inactivated state. The experimental temperature range of IFN-α2b “activation” is 36–37.5 °C, this interval corresponds to the internal temperature of the human body. It is also worth taking into account the following factor: in the range of 2–8 °C, which is characteristic of the storage conditions of the pharmaceutical, its emissivity is preserved. The speed of molecular motion at low temperatures will be lower than at high temperatures, but under these conditions it will slow down insignificantly [30,59]. When the medicines are heated to human body temperature, an increase in the radiothermal signal is observed, which can also be associated with an increase in the speed of molecular motion.

In the case of the VLP vaccine, activation was carried out using CreeLED-emitting LEDs with wavelengths of 412 nm. Some NPs are characterized by the induction of their properties through photocatalysis [60,61]. In the case of protein structure, the nitrogenous bases of nucleic acids are able to absorb blue light, which can lead to a change in the conformation of the protein structure. The influence of conformational changes on the emissivity is shown in Figure 2.

The ability to distinguish between medicines with different expiration dates and quality conditions plays an important role in the quality control of medications. Returning to Table 1 allows us to see how radiothermal emission results change for different IFNs and intranasal VLP vaccines for the prevention of COVID-19. It is quite difficult to visually identify an expired medicine; difficulties also arise in identifying spoiled pharmaceuticals. Modern pharmaceutical companies carry out expensive research in the area of calculating the approximate shelf life of medicines, but this does not prevent errors from occurring at the stage of their production or transportation [62]. Paying attention to the results obtained, it can be claimed that the obtained values are different. In the case of expired medicines, their intrinsic radiothermal emission is approximately twice reduced compared to the same medicine of appropriate quality, and for artificially aged pharmaceuticals, the indicators differ ~three–four times. The coagulation of the selected pharmaceuticals was confirmed by determining the size of the spectrum of NPs using the DLS method (Table 2). Despite the important difference between the emission values, there is no complete absence of emission, which may indicate the presence of activity in medicines that have expired [15]. The problem of the activity of pharmaceutical substances in preparations after the expiration date is open and currently being studied [63].

The main pharmacodynamic and pharmacokinetic parameters of any medication will depend on the choice of dosage form. The present-day pharmaceutical market for immunomodulatory pharmaceuticals, using IFNs as an example, offers various types of dosage forms: liquid, solid, soft, and gaseous [64]. When developing a method for controlling the quality of medications, an important goal was to understand the effect of different forms of the pharmaceutical on the conductivity of electromagnetic emission. Figure 3 shows the comparison of three different dosage forms of IFN-α2b. Despite the different dosages of the presented pharmaceuticals, the key difference in the detected radiothermal emission was their dosage forms. In the case of the liquid form, the highest emissivity was observed, while, for soft forms, the results were four times lower. It is assumed that the attraction of molecules in soft forms will be greater than in liquid forms; the particles are located more densely, which makes the process of molecule repulsion more difficult [30,31]. Describing this interaction between molecules, one can assume that the possibility of forming an electromagnetic field is not universal, but does not tend towards its complete absence either.

For each quality control method, it is necessary to detect the minimum possible concentration of the analyte. The limit of detection in the presented study was determined by measuring the intrinsic radiothermal emission from different volumes of IFN-α2b and different concentrations of VLP vaccines. The API content was recalculated for each selected volume. Starting from the first minimum possible detection volume of 1 μL (10IU), the emission values exceeded the control ones 20 times. The natural increase in the radiothermal signal from the measured volume can be explained by an increase in the interacting molecules of active substances, distributed throughout the thickness of the studied specimen. The mechanism of intermolecular interaction leading to the formation of plasma-like regions was described earlier. Interestingly, the kinetics of resonance energy transfer caused by plasmons is an open area for study [65]. The results of the study of the intra- and intermolecular transfer of vibrational energy in supramolecular structures state that the transfer of this energy occurs in a picosecond time scale [66]. The data obtained plays an important role both for the systematic view of the processes of intra- and intermolecular interactions, and for the scientific community in general.

The ability to study the intrinsic radiothermal emission not only in the laboratory using commercial IFNs, but also at the stages of preclinical studies of VLP vaccines, became the key focus in this presented study. Figure 6 shows the curve of the formation of the immune response in experimental individuals at different time stages of the study, based on the data obtained from their intrinsic radiothermal emission. The formation of an immune response to the administered vaccine, that is, the production of antibodies by the immune system, begins approximately on days 5–7 after vaccination [67,68]. Taking this fact into account, the results obtained can be interpreted as follows: on day 10, after the administration of the first component of the vaccine, a sharp spike in radiothermal emission is observed, which can be explained by the active response of the immune systems of the individuals. Literally a few days after re-vaccination, a sharp spike in radiothermal emission is observed, which may characterize the peak responses of the immune systems of the test subjects to the second component of the vaccine. The third stage of re-vaccination does not cause a visible increase in radiothermal emission, but rather, on the contrary, equalizes the parameters with the control group. Thus, the presented method for recording microwave radiothermal emission can be used not only for the quantitative quality control of IBPs in vitro, but also in detecting the activation of IFN synthesis in vivo.

## 4. Conclusions

Summarizing the results obtained, the main points can be highlighted. The described method provides an opportunity to control the radiothermal emission of medicines and IBPs without opening the primary packaging. There is a clear dependence of the radio signal on the selected dosage form of the pharmaceuticals (Figure 3). We have shown that it is possible to distinguish pharmaceuticals with different shelf lives using their intrinsic radiothermal emissions (proper shelf life, expired (more than 6 months ago), and artificially aged pharmaceuticals) (Table 1). The sensitivity of the proposed method makes it possible to identify IFNs in various volumes of aliquots and VLP vaccines with different concentrations (Figure 4). It is possible to use this method not only for the laboratory quality control of pharmaceuticals, but also at the preclinical trials to detect an immune response in living organisms in real time (Figure 6).

This study of the intrinsic radiothermal emission of medicines and IBPs is a promising area for the development of the latest express method for the quality control of medications. The method allows for the studying of pharmaceuticals without opening the primary package at any stage of their “life” and can also be further used in preclinical and clinical trials for the purpose of monitoring their effectiveness.

## Figures and Tables

**Figure 1 pharmaceutics-16-00180-f001:**
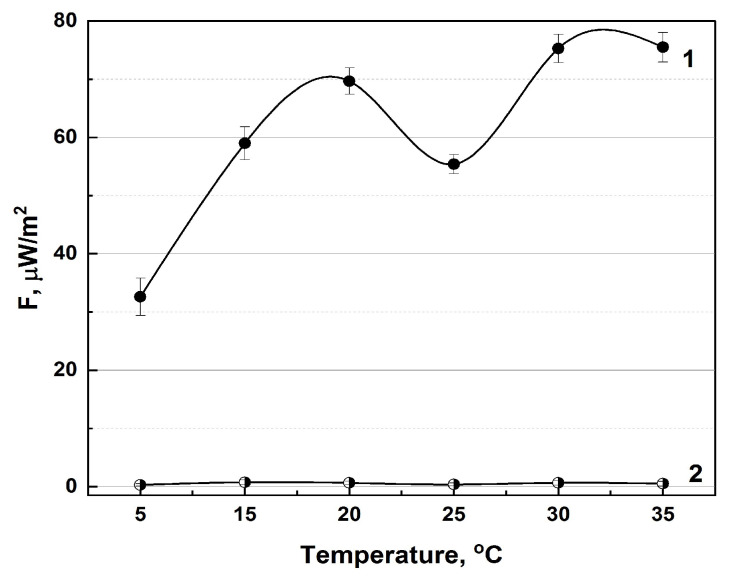
Dependence of the intrinsic radiothermal emission on temperature for the IFN—α2b medicines (**1**) in comparison with control values (**2**). Milli-Q high-resistivity water served as control (n = 7).

**Figure 2 pharmaceutics-16-00180-f002:**
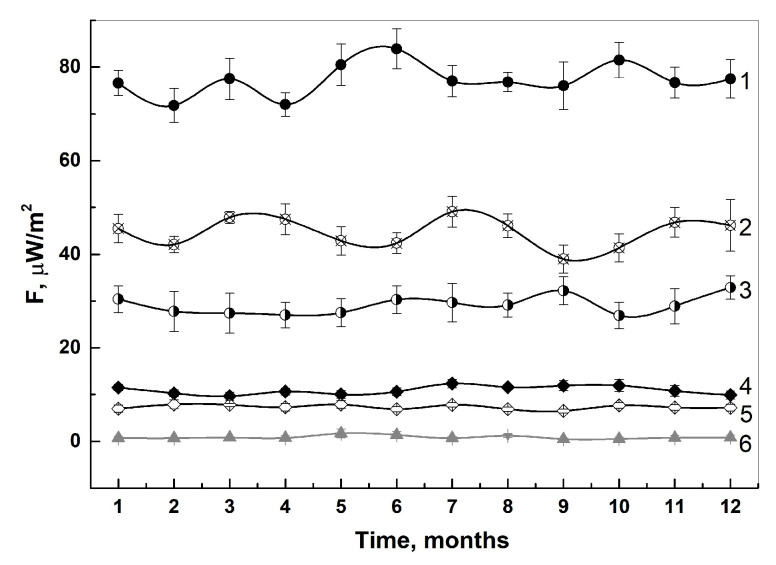
Annual time curve for assessing the repeatability of the results obtained when measuring intrinsic radiothermal emission of IBPs, where 1—IFN α-2b, 2—IFN-β-1a, 3—IFN-γ, 4—Gam-VLP-multivac, 5—Gam-VLP-rota, 6—control. All measurements were performed without opening the primary package (n = 7).

**Figure 3 pharmaceutics-16-00180-f003:**
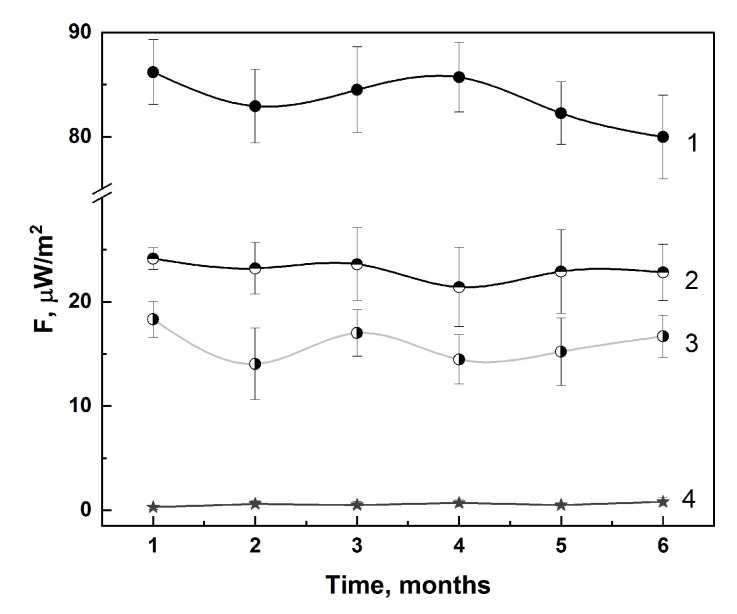
Comparison of the results of the intrinsic radiothermal emission of various dosage forms of IFN-α2b, where 1 is solution, 2 is ointment for external use, 3 is rectal suppositories, and 4 is control (n = 7).

**Figure 4 pharmaceutics-16-00180-f004:**
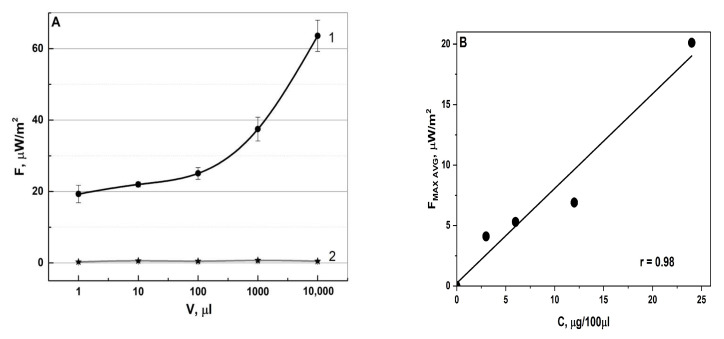
Dependence of the intrinsic radiothermal emission on the volume of the aliquot of IFN-α2b ((**A**), 1—IFN-α2b, 2—control) (n = 7), and VLP concentration at a fixed aliquot of 100 μL (**B**).

**Figure 5 pharmaceutics-16-00180-f005:**
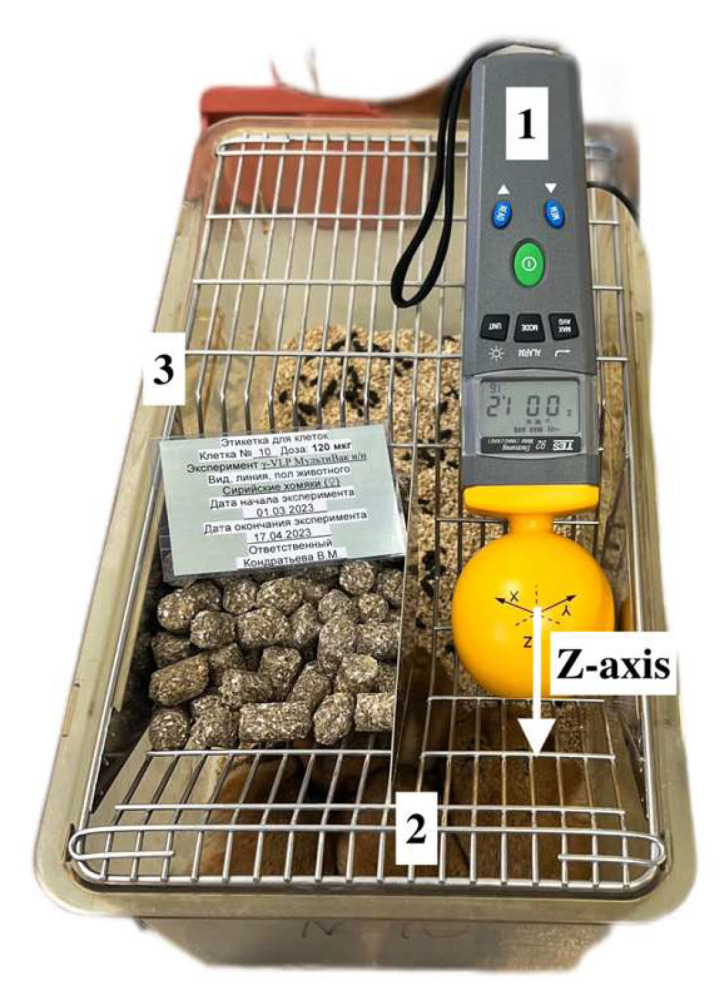
Geometry of the installation for measuring radiothermal emission, where 1—electromagnetic emission sensor TES-92; 2—immunized animals; and 3—cage for animals.

**Figure 6 pharmaceutics-16-00180-f006:**
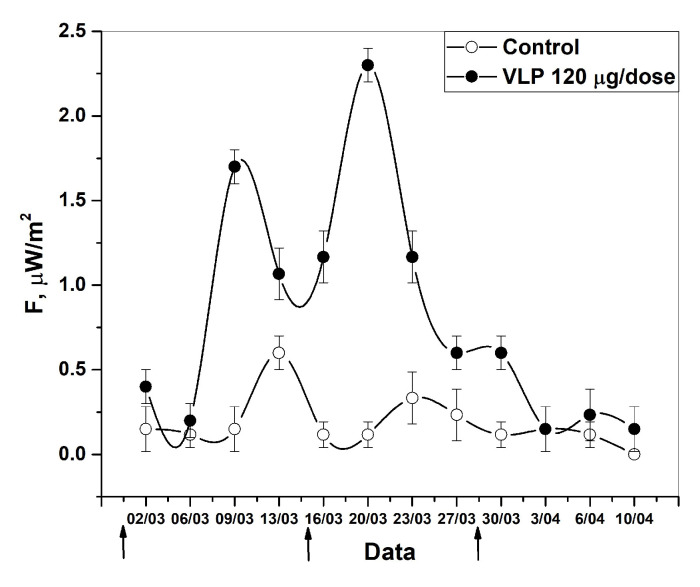
Control of the formation of an immune response in golden hamsters in response to the administration of a VLP vaccine by detecting radiothermal emission. The arrows indicate the days of vaccine administration (1st—01.03.23, 2nd—15.03.23, 3rd—29.03.23) (n = 7).

**Figure 7 pharmaceutics-16-00180-f007:**
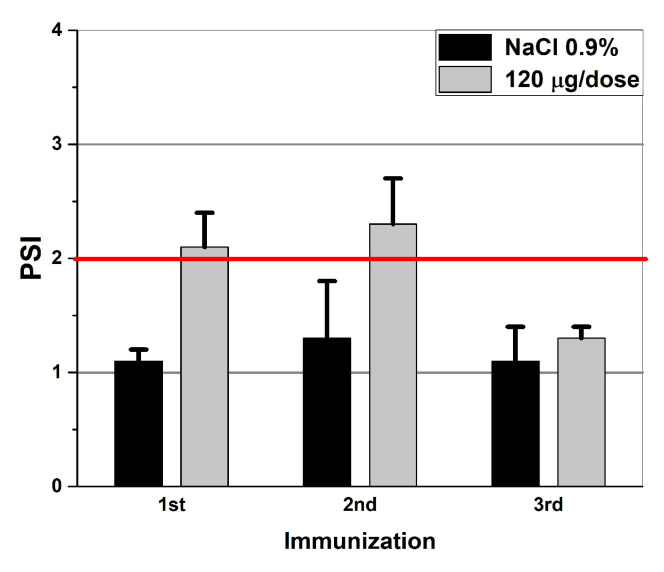
The reaction of the blast transformation of hamsters’ lymphocytes on day 10 after one, two, and three immunizations (n = 7).

**Table 1 pharmaceutics-16-00180-t001:** Comparison of the intrinsic radiothermal emissions of IBPs with various quality characteristics: proper shelf life, expired, and artificially aged therapeutics.

Specimens	IFN-α2b	IFN-β1a	IFN-γ	VLP-SARS
	F ± Sr, μW/m^2^			
Proper shelf life	89 ± 4	41 ± 3	31 ± 3	13 ± 1
Expired	55 ± 3	21 ± 3	19 ± 3	9 ± 2
Artificially aged	31 ± 4	19 ± 3	20 ± 3	4 ± 1

**Table 2 pharmaceutics-16-00180-t002:** Comparison of size spectra of IBPs with various quality characteristics: proper shelf life and artificially aged pharmaceuticals.

Specimens	IFN-α2b	IFN-β1a	IFN-γ	VLP-SARS
	D ± Sr, nm			
Proper shelf life	190 ± 37	220 ± 40	295 ± 65	190 ± 22
Artificially aged	615 ± 71	531 ± 68	712 ± 60	220 ± 31

## Data Availability

The original contributions presented in the study are included in the article, further inquiries can be directed to the corresponding author.

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
