# Peer review of "Controlling the Quality of Nanodrugs According to Their New Property—Radiothermal Emission"

_pharmaceutics, 2024, doi:10.3390/pharmaceutics16020180_

Round 1

Reviewer 1 Report

Comments and Suggestions for Authors

Abstract

Abstract is generally well written.

Keywords

“Quality control of pharmaceuticals with nanoparticles” is too long for a keyword, separate into e.g., quality control, nanoparticle.

Introduction

Introduction is clear, well written. Aims of the study are defined properly.

Lines 106 to 116 can be rephased to better introduce the aim of this current study. It still feels like a literature introduction section instead of almost point-by-point-like aim determination.

Materials and methods

Section 2.2. A brief summary of the referenced papers and the synthesizes can enhance the quality  of the manuscript, if of course, possible.

All other methods are clear and relevant to the aims of the study.

Results

Figures are in excellent quality. Figure captions are clear.

Figure 4. months instead of monthes.

Results are also clear and the description parts are understandable and coherent with the aims and the methods previously discussed.

Discussion and conclusions

Well written.

All in all, the manuscript is of good quality, and I recommend it for publication after some minor changes and spell checking.

Author Response

1) Abstract

Abstract is generally well written.

Answer: Thank You very much for Your comment on the abstract!

2) Keywords

“Quality control of pharmaceuticals with nanoparticles” is too long for a keyword, separate into e.g., quality control, nanoparticle.

Answer: Thank you very much, the long keyword has been replaced with the ones You suggested.

3) Introduction

3.1 Introduction is clear, well written. Aims of the study are defined properly. 

Answer: Thank You very much!

3.2 Lines 106 to 116 can be rephased to better introduce the aim of this current study. It still feels like a literature introduction section instead of almost point-by-point-like aim determination.

Answer: The introduction section has been rewritten due to requests from reviewers. The aim is clearly reflected in the Introduction.

4) Materials and methods

4.1 Section 2.2. A brief summary of the referenced papers and the synthesizes can enhance the quality of the manuscript, if of course, possible.

Answer: Lines 95-106 show the methods used for the synthesis and quality control of VLP-Rota. For VLP SARS, a link is provided that discloses the necessary information.

4.2 All other methods are clear and relevant to the aims of the study.

Answer: Thank You very much!

5) Results

5.1 Figures are in excellent quality. Figure captions are clear.

Answer: Thank You very much!

5.2 Figure 4. months instead of monthes.

Answer: it has been fixed in figure 2 and figure 3 (ex-figure 4)

5.3 Results are also clear and the description parts are understandable and coherent with the aims and the methods previously discussed.

Answer: Thank You very much!

6) Discussion and conclusions

Well written.

Answer: Thank You very much!

All in all, the manuscript is of good quality, and I recommend it for publication after some minor changes and spell checking.

Answer: Thank you very much for your work and positive feedback about our work!

Reviewer 2 Report

Comments and Suggestions for Authors

The abstract should be devoid of undefined acronyms. It must containing some numerical values of the results. Please add concise conclusion at the end of abstract.

Please check the manuscript for minor grammar mistakes.

Please, rewrite the introduction section as one synchronized unit. Avoid the use of long sentences or paragraphs without references citation . Also, add the aim of the study in a clear sentences at the end of introduction section.

Please, check the use of acronyms throughout the manuscript. No need single or minimal use of words.

What the type and number of animals used in this study? What about the ethical approval of this work? What is the basis of doses selection? How the animals euthanization was achieved.

Please, add paragraph at the end of materials and methods about statistically analysis including software used in analysis of data, data expression, p values.

Each figure and table should be self presentable in the term of acronyms, samples number, data expression, ect.

Please, refine ths discussion section for long sentences and paragraphs without references citation. Also, support your results with similar findings.

The reference list must updated, including 2024 in the reference list.

Please, rewrite the conclusion as a single synchronized paragraphs.

Comments on the Quality of English Language

Please, check the manuscript for minor grammar editing, syntax and sentences structure.

Author Response

1) The abstract should be devoid of undefined acronyms. It must containing some numerical values of the results. Please add concise conclusion at the end of abstract.

Answer: Acronyms have been introduced throughout the abstract and numerical values of the results.

2) Please check the manuscript for minor grammar mistakes.

Answer: It has been verified.

3) Please, rewrite the introduction section as one synchronized unit. Avoid the use of long sentences or paragraphs without references citation . Also, add the aim of the study in a clear sentences at the end of introduction section.

Answer: The introduction section has been rewritten as one synchronized unit. The aim is clearly reflected in the Introduction.

4) Please, check the use of acronyms throughout the manuscript. No need single or minimal use of words.

Answer: Acronyms have been checked and superfluous ones have been removed, replaced with the full word

5) What the type and number of animals used in this study? What about the ethical approval of this work? What is the basis of doses selection? How the animals euthanization was achieved.

Answer: Lines 151-168. The answers to the above questions are given.

6) Please, add paragraph at the end of materials and methods about statistically analysis including software used in analysis of data, data expression, p values.

Answer: Lines 197-200. Paragraph 2.6. Metrological support was introduced for the program used during the processing of the results.

7) Each figure and table should be self presentable in the term of acronyms, samples number, data expression, ect.

Answer: we believe that each of the figures and tables indicated in the article are self-representative

8) Please, refine ths discussion section for long sentences and paragraphs without references citation. Also, support your results with similar findings.

Answer: we reinforced the “Discussions" section with links in those sentences and paragraphs where possible. Links 15,30,31,64 (highlighted in red).

9) The reference list must updated, including 2024 in the reference list.

Answer: References to the 2024 article, reference number 15, have been added to the list of references.

10) Please, rewrite the conclusion as a single synchronized paragraphs.

Answer: Lines 465-481. The conclusions are presented in separate synchronized paragraphs.

Reviewer 3 Report

Comments and Suggestions for Authors

Comments to the authors:

The work described in the present manuscript is consistent with the scope of the journal.

Authors proposed a new method for quality control of immunobiological preparations based on the detection of their intrinsic radio-thermal emission as well as the applicability of the methodology in in vivo studies. Moreover, the ability to control radio-thermal emission without opening the original package was achieved, which represent a great advantage of the methodology.

Nevertheless, major reviews are required prior to a possible publication. For this, some points must be addressed, as described below:

Major comments:

1. Line 12: explain what NPs emit

2. Introduction is too long, please reduce it, prioritizing the most relevance information.

3. Lines 272-276: please mention herein table 2 that supports the described results. Figure 3 is redundant since the same information is described in table 2.

4. Line 294-300: a refence is needed to support his affirmation or otherwise declare the speculative nature of these sentence.

5. Figure 7: there is no indication on the number of independent experiments performed as well as the obtained error bar. Please include this information.

6. Figure 8: there is no indication on the number of independent experiments performed. Please include this information.

7. As a general comment, in the section 3 (Results), authors discussed the obtained results, which are then debated in more detailed in section 4. I believe that both sections can be combined, and section 3 can be designated as “3- results and discussion” since it is more realistic.

Minor comments:

  1. Line 16: replace 24.106 by 24x106. The same in lines 75, 124-126
  2. Line 17: define VLP
  3. Line 35: define ARVI
  4. Line 67: replace 3.106 IU by 3x106 IU
  5. Uniformize the word “radiothermal” (must be radio-thermal) along all the manuscript.
  6. Line 324 and 477: in vivo must be in italic
  7. Line 427: replace Table 3 by Table 2
Comments on the Quality of English Language

Minor editing of English language required

Author Response

Major comments:

  1. Line 12: explain what NPs emit

Answer: Line 12. “Previous studies have shown that nanoparticles of complex shape have their intrinsic radiothermal emission in the millimeter range”.

  1. Introduction is too long, please reduce it, prioritizing the most relevance information.

Answer: The introduction section has been rewritten including the most relevance information. The aim is clearly reflected in the Introduction.

  1. Lines 272-276: please mention herein table 2 that supports the described results. Figure 3 is redundant since the same information is described in table 2.

Answer: Line 267. A reference to table 2 has been introduced. Figure 3 has been removed, the numbering of subsequent figures has been corrected.

  1. Line 294-300: a refence is needed to support his affirmation or otherwise declare the speculative nature of these sentence.

Answer: Lines 282-288. Added links 30, 31.

  1. Figure 7: there is no indication on the number of independent experiments performed as well as the obtained error bar. Please include this information.

Answer: Line 348. Standard deviations for Figure 6 have been added. The numbering of the figures has changed, due to the deletion of Figure 3, now this figure is numbered 6.

  1. Figure 8: there is no indication on the number of independent experiments performed. Please include this information.

Answer: The number of independent measurements performed has been added to Figure 7 (ex-Figure 8).

  1. As a general comment, in the section 3 (Results), authors discussed the obtained results, which are then debated in more detailed in section 4. I believe that both sections can be combined, and section 3 can be designated as “3- results and discussion” since it is more realistic.

Answer: Line 201. We combined the results section and the discussion section into a single section "results and discussion"

Minor comments:

1) Line 16: replace 24.106 by 24x106. The same in lines 75, 124-126

Answer: Line 16, 68, 91-93 The multiplication sign X has been applied.

2) Line 17: define VLP

Answer: The VLP definition has been introduced.

3) Line 35: define ARVI

Answer: The ARVI abbreviation has been removed

4) Line 67: replace 3.106 IU by 3x106 IU

Answer: Line 68 The multiplication sign X has been applied.

5) Uniformize the word “radiothermal” (must be radio-thermal) along all the manuscript

Answer: unified, radiothermal is written together link: https://doi.org/10.3390/pharmaceutics15030966

6) Line 324 and 477: in vivo must be in italic

Answer: Line 159, 312, 464, 490 in vivo written in italics

7) Line 427: replace Table 3 by Table 2

Answer: Line 267, 414 It has been replaced

Round 2

Reviewer 3 Report

Comments and Suggestions for Authors

The authors have addressed my concerns in the present revision. The modifications performed in the manuscript clarified most of the non-well explained topics and significantly improved the quality of the manuscript. So, I consider the paper suitable for publication in the present form.